

**Morphological plasticity of root growth under mild water stress**
**increases water use efficiency without reducing yield in maize**
Qian Cai[1,2], Yulong Zhang[1*],Zhanxiang Sun[2*], Jiaming Zheng[2],Wei Bai[2], Yang Liu[2]
Liangshan Feng[2], Chen Feng[2], Zhe Zhang[2], Ning Yang[2], Jochem B. Evers[4], Lizhen
Zhang[3]
[1]College of Land and Environment, Shenyang Agricultural University, Shenyang,
110161, Liaoning, China;
[2]Tillage and Cultivation Research Institute, Liaoning Academy of Agricultural
Sciences, Shenyang, 110161, Liaoning, China;
[3]College of Resources and Environmental Sciences, China Agricultural University,
Beijing, 100193, China.
[4]Wageningen University, Centre for Crop Systems Analysis (CSA),
Droevendaalsesteeg 1, 6708 PB Wageningen, The Netherlands
**First author**
Qian Cai, Email: caiqian2005@163.com
**\*Corresponding author**
Yulong Zhang, Email: ylzsau@163.com; Zhanxiang Sun, Email:
sunzhanxiang@sohu.com





**Abstract**
There is a significant potential to increase yield of maize (*Zea mays* L.), a global
major crop, in rain-fed condition in semi-arid regions, since the large yield gap is
mainly caused by frequent droughts halfway the crop growing period due to uneven
distribution of rainfall. It is questionable if irrigation systems are economically
required in such a region since total amount of rainfall generally meet the crop
requirement. This study therefore aimed to quantitatively determine the effects of
water stress during jointing to filling stages on root and shoot growth and the
consequences for maize grain yield, above- and below-ground dry matter, water
uptake (WU) and water use efficiency (WUE). Pot experiments were conducted in
2014 and 2015 with a mobile rain shelter. The experiments consisted of three
treatments: (1) no water stress; (2) mild water stress; and (3) severe water stress.
Maize yield in mild water stress across two year was not significantly affected, while
severe stress reduced yield by 56 %. Water stress decreased root biomass slightly but
shoot biomass substantially. Mild water stress decreased root length but increased root
diameter, resulting a no effect on root surface area. WU under water stress was
decreased, while WUE for maize above-ground dry matter under mild water stress
was increased by 20 % across all years, and 16 % for grain yield WUE. Our results
demonstrates that irrigation systems in studied region might be not economically
necessary because the mild water stress does not reduce crop yield. The study helps to
understand crop responses to water stress during critical water-sensitive period and to
mitigate drought risk in dry land agriculture.
**Keywords:** root diameter; root length; root surface area; root/shoot ratio; yield
components; water utilization



## 1. Introduction


Maize (*Zea mays* L.) as one of the most important crops globally, is a major food
crop in northeast China with an average yield of around 5.3 t ha$^{-1}$ (Dong et al., 2017).
However, the yield gap to the potential of 10.9 t ha$^{-1}$ is still large (Liu et al., 2012),
mainly due to lack of irrigation and frequent summer droughts caused by an uneven
distribution of rainfall during the crop growing season. As global warming is
anticipated to cause a higher expected frequency of extreme climate events (IPCC,
2007), drought risk for agricultural production in this region is likely to increase
(Song et al., 2014; Yu et al., 2014). Water stress changes crop response in
morphological and physiological traits (Pampino et al., 2006). Warming and dry
trends under climate change would result deleterious effects on crop photosynthesis
and yield (Richards, 2000).
Although the total amount of rainfall can meet the requirement of rain-fed maize
in the semi-arid northeast China, the yearly and seasonal variation often causes a
frequent drought (mostly mild water stress) during summer and results in high risk of
yield loss. It can be questioned whether irrigation systems are economically required
in this situation, since it is not quantitatively known how the crop yield and water use
efficiency would be affected by such drought stress during summer in this region.
Suppression of yield by water stress is caused by reducing crop growth (Payero
et al., 2006), canopy height (Traore et al., 2000), leaf area index (NeSmith and Ritchie,
1992) and root growth (Gavloski et al., 1992). Crop shoot development and biomass
accumulation are greatly reduced by soil water deficit at seeding stage (Kang et al.,
2000). Short-duration water deficits during the rapid vegetative growth period causes
around 30% loss in final dry matter (Cakir, 2004). The reduction of maize yield by
water stress is caused by decreases in yield components such as ear size, number of





kernel per ear and/or kernel weight (Ge et al., 2012), especially during or before
maize silk and pollination period (Claassen and Shaw, 1970). The accumulative
biomass and harvest index (the ratio of grain yield over total aboveground dry matter)
are decreased under water stress during anthesis (Traore et al., 2000).
Water use efficiency (WUE, expressed in kg yield obtained per $m^3$ of water) is
notably reduced by severe water stress especially at vegetative and reproductive
stages. Interestingly a moderate water stress at V16 and R1 stages in maize increased
WUE (Ge et al., 2012) because it did not significantly affected the ecophysiological
characteristics during vegetative stages. The irrigation deficits before the maize
tasseling stage are often used for improving WUE in regions with serious water
scarcity, e.g. North China Plain (Qiu et al., 2008; Zhang et al., 2017). For example, in
winter wheat WUE was increased continuously from 1987 to 2015 especially under
water stress condition that was obtained from a increased harvest index and the
reduced soil evaporation (Zhang et al., 2017). Under water stress, plant
photosynthesis and transpiration decreases due to a decrease in stomata conductance
(Killi et al., 2017) which is induced by increased concentration of abscisic acid (ABA)
in plant (Beis and Patakas, 2015). However, limited acknowledge exists on how much
the assimilate partitioning between shoot and root in maize is affected by water stress
during middle and late growing stages, and if the root growth regulated by water
stress could improve maize yielding and water use efficiency.
Since field water stress experiments were difficult to carry out in rain-fed
agriculture, a large mobile rain shelter was used in this study to control water stress
(NeSmith and Ritchie 1992). The objective of this study was to quantify maize shoot
and root growth, grain yield and WUE under different water stresses, to understand
the crop response to water stress during critical water-sensitive period.




## 2. Materials and methods

### 2.1. Experimental design

The experiments were conducted in Shenyang (41°48′N, 123°23′E), Liaoning
province, northeast China in 2014 and 2015. The experimental site is 45 m above sea
level. Annual potential evaporation is 1445 mm, total precipitation is 720 mm, and
mean air temperature is 8 °C. The average length of the frost-free period is 150-170
days. Average relative humidity is 63 %. Annual wind speed is 3.1 m s$^{-1}$. The climate
is a typical continental monsoon climate with four distinct seasons, characterized as a
hot summer and cold winter. Total rainfall during crop growing season (May to
September) was 295 mm in 2014 and 436 mm in 2015. The annual mean air
temperature was 9.5 °C in 2014 and 9.1 °C in 2015. The mean air temperature during
crop growing season was 20.2 °C in 2014 and 19.4 °C in 2015 (Fig. 1).
Maize plants were grown in pots in three treatments: (1) no water stress; (2) mild
water stress and (3) severe water stress. The water supply was controlled by a mobile
rain shelter. The shelter was moved away from the experimental plots in no rain days
and covered before a rain came, therefore the effect of shelter on incoming radiation
could be ignored. Water treatments began from maize jointing (V6, with 6 fully
expended leaves) to filling stages (R3, milk) (Abendroth et al., 2011). Supplement
water was given once per 5 days before starting water treatment with same amount for
all pots, and once per 3 days during the period of water treatments. The detail amount
of water supplied to each treatment was listed in Table 1. The experiments entailed a
completely randomized block design with three replicates. Each treatment consisted
of 12 pots (one plant per pot) and divided into 3 replicates (4 pots each). At each
sampling time (totally sampling 4 times), one pot was used.





Each pot was 40 cm in diameter and 50 cm in height, filled with 40 kg naturally
dried soil with a bulk density of 1.31 g cm$^{-3}$. The soil was sandy loam with a pH of
6.15, total N of 1.46 g kg$^{-1}$, total of P 0.46 g kg$^{-1}$ and total K of 12.96 g kg$^{-1}$. 46.5 g
compound fertilizer (N 15 %, $P_2O_5$ 15 % and $K_2O$ 15 %) and 15.5 g diammonium
phosphate (N 18 % and $P_2O_5$ 46 %) were applied to each pot before sowing. There
was no other fertilizer applied during maize growing season. Maize cultivar used in
both years was Liaodan 565, a locally common used drought-resistant cultivar. One
plant was grown in each pot. Maize was sown on 13-May and harvested on 30-Sept in
both 2014 and 2015.

**2.2 Dry matter and grain yield measurements**

To determine maize dry matter, four plants were harvested on 49 (V6), 77 (VT),
(R3) and 141 (R5) days after sowing (DAS) in 2014, and only one sampling was
done on 132 DAS in 2015. The samples were separated into root and shoot, and dried
in an oven at 80 $^{o}$C for 48 hours until reaching a constant weight. The shoot/root ratio
was calculated using dry matters measured.
Grain yield was measured by harvesting all cobs in a pot in maize harvesting
time. The grain was sundried with a water content of 15%. Yield components i.e. ear
(cob) numbers per plant, kernel numbers per ear and thousand kernel weight was
measured for each plant.

**2.3 Root measurements**

Root growth and morphological traits (root length, diameter and surface area)
were measured four times during crop growing season on 49 , 77, 113, 141 DAS only
in 2014. The whole roots were collected per pot at the time of dry matter





measurements. Root samples were carefully washed with tap water to remove
impurities. The cleaned roots were placed on a glass plate of a root system scanner.
Scanned root images were analyzed by a plant root image analyzer WinRHIZO PRO
2009 (Regent Instruments Inc., Canada) to quantify total root length (m), diameter
(mm) and surface area ($m^2$) per plant (pot).

**2.4 Measuring soil moisture content, water uptake and water use efficiency**
Soil moisture contents were measured by a soil auger at sowing and harvesting
times. Soil cores were taken from the middle of a pot for each 10 cm layer. After
measuring fresh soil weight, soil samples were dried in an oven at 105 $^o$C for around
48 hours until a constant weight was reached. The gravimetric soil moisture contents
(%, g $g^{-1}$) measured by soil auger were calculated into volumetric soil moisture
content (%, $m^3$ $m^{-3}$) by multiplying with soil bulk density.
Water uptake (WU) of maize was calculated using a simplified soil water balance
equation (Kang et al., 2002). Because the pot experiments were sheltered, rainfall,
drainage and capillary rise of water did not occur in this situation and therefore were
not taken into account in the calculation of WU:
$$WU = I + \Delta S \tag{1}$$
where WU (mm) is crop water uptake (mm) during whole crop growing season, I is
the amount of water supplied to each pot (mm). ΔS is the changes of soil water
amount between sowing to harvesting dates.
Water use efficiency (WUE) was calculated by measured final yield or
above-ground dry matter (shoot) and total WU during crop growing season (Zhang et
al., 2007).
$$WUE = Y/WU \tag{2}$$

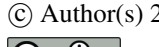



where WUE (g m$^{-2}$ mm$^{-1}$ or kg m$^{-3}$) is water use efficiency expressed in gain yield
WUE$_Y$ or dry matter WUE$_{DM}$. Y (g m$^{-2}$) is grain yield or dry matter. WU (mm) is total
water uptake during maize growing season.

**2.5. Statistical analysis**

Analysis of variance on yield, WU, WUE, and dry matter for shoot and root were

performed using General Linear Model of SPSS 20 (SPSS Inc., Chicago, USA). The
differences between means were evaluated through LSD multiple comparison tests at
a significant level of 0.05.

**3. Results**
**3.1 Yield and yield components**

Maize yield in mild water stress across two year was not significantly different

with no stress control, while that in severe stress was 56 % lower (Table 2). The
decrease of maize yield in severe water treatments was due to the decreases in ear
number, kernel number and harvest index (HI). However, water stress did not affect
kernel weight, while other yield components were decreased. Year effect was only
significant for HI, which was likely caused by the variation in air temperature: the
cooler weather in 2015 during maize growing season decreased HI comparing with a
warmer 2014. There were no significant interactions between year and treatment.

**3.2 Above- and below-ground dry matters**

Mild water stress did not reduce root dry matter (Fig. 2 a, b), but greatly reduced

shoot dry matter, especially at grain filling stage (113 DAS) (Fig. 2 c, d). The severe
water stress decreased both root and shoot dry matter compared with no stress control,





but the magnitude of the decrease in shoot was much larger than in root. At maize VT
stage (77 DAS), as roots generally reach their maximum size, root dry matter under
severe water stress was much lower than mild and no water stress treatments.
However, it became less different later in the season, which indicated a strong
complementarily growth of root system during water stress. Due to the different
responses of shoot and root to water stress, the root/shoot ratios under water stress
were increased (Fig. 2 e, f), especially during crop rapid growing period (77 to 113
DAS).

**3.3 Root length, diameter and total surface area per pot affected by water stress**
Root length per plant was much lower under severe water stress, especially at VT
stage (77 DAS). Mild water stress during maize middle growing season also
decreased root length, but the difference with no stress control was much smaller than
severe stress (Fig. 3 a). Root diameter under both mild and severe water stress
treatments was much higher comparing with no stress control (Fig. 3 b), especially at
late growing season. The decrease in root length under water stress was partially
compensated by the increase in root diameter. This resulted in a small change in total
root surface area (Fig. 3 c), especially during maize reproductive growth period (113
DAS).

**3.4 Water uptake and use efficiency**
Total water uptakes (WU) under water stress treatments were lower than under
no stress control (Fig. 4). Water use efficiency for maize above-ground dry matter
($WUE_{DM}$) under water stress was increased 30.3 % comparing with no stress control,
across all years and treatments (Fig. 4 b). The $WUE_{DM}$ in severe water stress was the





highest. However, WUE for grain yield in severe water stress was not significantly
different with that in the control, while that in mild water stress showed a increase
(15.7 %) across two years (Fig. 4 c). The difference between WUE in dry matter and
grain yield was due to a significant decrease in HI under severe water stress (Table 2).

**4. Discussion and conclusions**

Mild water stress during the middle growing period did not significantly

suppress grain yield. It is different with previous report that maize yield is much more
affected by water stress during flowering stage than other stages (Doorenbos et al.,
1979), probably due to the ecological conditions and drought-sensitivity of cultivars.
Mild water stress reduced total water uptake, resulting a 20 % higher WUE in dry
matter production and a 16% higher WUE in yield. The increase in WUE under mild
water stress was partially from the different responses of shoot and root growth to
water stress, resulting in an increase in root/shoot ratio. The water stress before
flowering reduced root growth, however, this reduction was compensated for later by
complementarily lateral root growth.

Severe water stress greatly reduced both shoot and root biomass, which was due

to a large decrease in water uptake. Canopy transpiration is largely determined by net
radiation absorption by the leaves in the canopy (Monteith, 1981). Large decreases in
shoot growth, i.e. less biomass and leaf area, reduces the light interception. Under
mild water stress during vegetative and tasselling stages, the shoot growth was
reduced in this study and previous report, e.g. plant height, leaf area development
(Cakir, 2004), however, mild soil water deficit may also reduce water loss from plants
through physiological regulation (Davies and Zhang, 1991). A moderate soil drying at
the vegetative stage encourages root growth and distributing in deep soil (Jupp and



Newman, 1987; Zhang and Davies, 1989), which is consistent with our findings.
Large root system with deep distribution is beneficial for water-limited agriculture
(McIntyre et al., 1995).
We found an increase in root diameter under water stress, although root length
was decreased. This result indicated that the lateral roots under water stress were
probably less than under no water stress. That may limit water absorption since the
lateral roots is younger and more active in uptake function (Lynch, 1995). Average
root diameters in all treatments decreased from 77 to 113 DAS, which was caused by
highly emerged lateral roots after the main root system reached its maximum (VT
stage). The higher average root diameter in water stress treatments than in the control
at 141 DAS was probably due to a fast senescence of late developed lateral roots
under water stress.
Our results on root morphological plasticity affected by water stress provided
another evidence for enhancing WUE and maintaining yielding by a mild water
deficit. However, the mechanism that determines crop response to water stress may
also involve other processes, e.g. intercellular $CO_2$, stomatal conductance,
photosynthetic rate, oxidative stress, sugar signaling, membrane stability and root
chemical signals (Xue et al., 2006; Dodd, 2009). The relationship between carbon
assimilation and water loss in relation to the assimilates between reproductive and
vegetative organs responding to soil water availability have been widely explored to
understand the physiological mechanism of improving WUE under moderate water
stress (Ennahli and Earl, 2005; Xue et al., 2006; Zhang et al., 2013). Under water
limitation, photosynthesis and transpiration rates are in a permanent tradeoff
regulating by stomata conductance. The abscisic acid (ABA)-based drought stress
chemical signals regulates crop vegetative and reproductive development and





contributes to crop drought adaptation (Killi et al., 2017). Increased concentration of
ABA in the root induced by soil drying may maintain root growth and increase root
hydraulic conductivity, thus increases crop water uptake and thereby postpone the
development of water deficit in the shoot (Liu et al., 2005). The increase of ABA
induces stomatal closure and reduces crop transpiration (Haworth et al. 2016), net
photosynthesis and crop growth (Killi et al., 2017).

Mild water stress during middle crop growing period could potentially maintain

maize yield and substantially reduced the water consumption at the same time. Thus,
the water use efficiency was increased by water deficit (Liu et al., 2016). However,
the maintenance of crop growth under water stress was limited by the severity of the
stress. Under severe water stress, maize growth failed to be compensated by structural
and functional plasticity in plant growth. Our result differed from a previous study,
which showed mild water stress also seriously affected crop production (Kang et al.,
2000). This is likely due to our choice for a drought-resistant variety (Zhengdan 565)
and the difference in ecological zones. Genotype-dependent relationships between
yield and crop growth rate would be stronger under water stress than under no stress
condition (Lake and Sadras, 2016).

The maize yield in 2015 was much lower than in 2014 independent of water

stress. That might be caused by a higher maximum air temperature in 2015 (32.0 $^{\circ}$C)
than in 2014 (29.1 $^{\circ}$C) during flowering period. High air temperature would reduce
maize pollination (Muller and Rieu, 2016) and directly affected yield formation and
HI.

This study clearly demonstrates that the maize yield under mild water stress

during summer does not decrease but the water use efficiency would increase due to
changes in root and shoot growth. A higher root/shoot ratio under mild water stress




allows plant efficiently use limited soil water. In rain-fed maize production in a region
with frequent drought, to optimizing maize yield, the agronomic managements, e.g.
cultivar selection, adjusting sowing windows (Liu et al., 2013; Lu et al., 2017) and
ridge and furrow cultivation (Dong et al., 2017) could be applied. Our study provides
interesting evidences to understand crop responses to water stress, especially on root
morphological plasticity in a drought environment. The results could be further
applied combining with crop model (Mao et al., 2015) to mitigate climate risk (e.g.
drought) in dry land agriculture globally.

**Author contribution**

Z. Sun, Y. Zhang, J. Zheng and Q. Cai conceived and designed the experiments.

Q. Cai, W. Bai, Y. Liu, L. Feng, C. Feng, Z. Zhang and N. Yang performed the
experiments. L. Zhang, Q. Cai and J.B. Evers analyzed the data and wrote the paper.

**Competing interests**

The authors declare that they have no conflict of interest.


**Special issue statement**

Special issue: Ecosystem processes and functioning across current and future

dryness gradients in arid and semi-arid lands.

**Acknowledgements**

This research was supported by the National key research and development

program of China(2016YFD0300204), the International Cooperation and Exchange
(31461143025) and the Youth Fund (31501269) of the National Science Foundation





of China, Liaoning BaiQianWan Talent Program (201746), Outstanding Young
Scholars of National High-level Talent Special Support Program of China.

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



**Table 1** Water treatments during crop growing seasons in 2014 to 2015.

| Year | Water treatment | Initial volumetric soil moisture content (%) | Actual water supply at three growing periods (mm) | | | |
| | | | Early (16-29 DAS[1]) | Middle (30-102 DAS) | Late (103-121 DAS) | Total |
|---|---|---|---|---|---|---|
| 2014 | No stress | 24.4 | 11.9 | 478 | 56 | 545 |
| | Mild stress | 24.8 | 11.9 | 299 | 56 | 366 |
| | Severe stress | 24.9 | 11.9 | 122 | 56 | 190 |
| 2015 | No stress | 25.3 | 11.9 | 510 | 32 | 553 |
| | Mild stress | 25.3 | 11.9 | 334 | 32 | 378 |
| | Severe stress | 24.4 | 11.9 | 159 | 32 | 203 |

[1]DAS refers days after maize sowing.






**Table 2** Yield and yield components affected by different water stresses in 2014 to

442 2015

| Year | Water treatment | Ear number | Kernel number | Thousand kernel weight | Yield per plant | Harvest index |
|---|---|---|---|---|---|---|
| | | ear plant⁻¹ | kernel ear⁻¹ | g | g plant⁻¹ | g g⁻¹ |
| 2014 | No stress | 2.0±0.0a | 354±32a | 440±6.8a | 301±33a | 0.36±0.01a |
| | Mild stress | 2.0±0.0a | 350±16a | 416±1.2b | 276±14a | 0.37±0.01a |
| | Severe stress | 2.0±0.0a | 245±35b | 412±3.7b | 166±25b | 0.27±0.02b |
| 2015 | No stress | 2.0±0.0a | 341±67a | 426±12a | 240±60a | 0.29±0.04a |
| | Mild stress | 2.0±0.0a | 244±53a | 427±22a | 168±42ab | 0.25±0.03a |
| | Severe stress | 1.3±0.3b | 172±46a | 412±16a | 81±22b | 0.17±0.04a |
| mean | No stress | 2.0±0.2a | 347±38a | 432±7.5a | 266±36a | 0.32±0.03a |
| | Mild stress | 2.0±0.0a | 289±36ab | 422±12a | 214±32a | 0.30±0.03ab |
| | Severe stress | 1.6±0.0b | 203±31b | 412±8.5a | 118±23b | 0.21±0.03b |
| P | Treatment | 0.021 | 0.003 | 0.556 | 0.005 | 0.013 |
| | Year | 0.184 | 0.514 | 0.889 | 0.237 | 0.039 |
| | Treat×Year | 0.111 | 0.664 | 0.555 | 0.835 | 0.758 |

Same small letters indicate no significant difference between water treatment within same year at $a$=0.05.




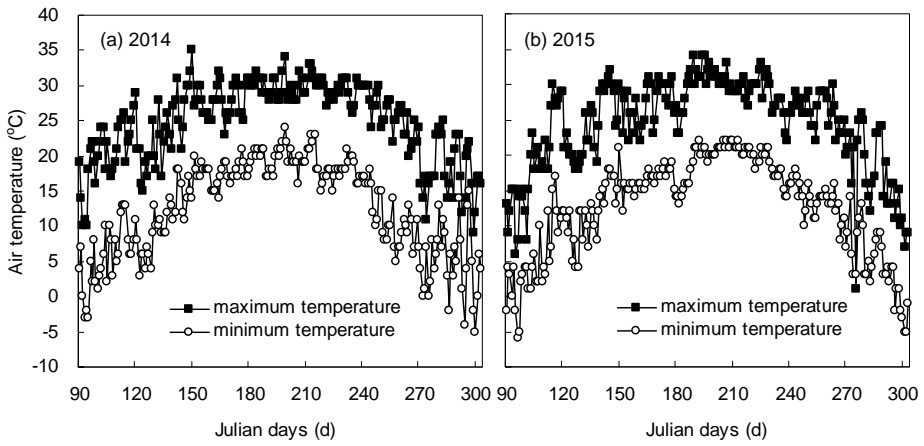


**Fig. 1** Daily maximum and minimum air temperatures in 2014 and 2015 in
Shengyang, Liaoning, China




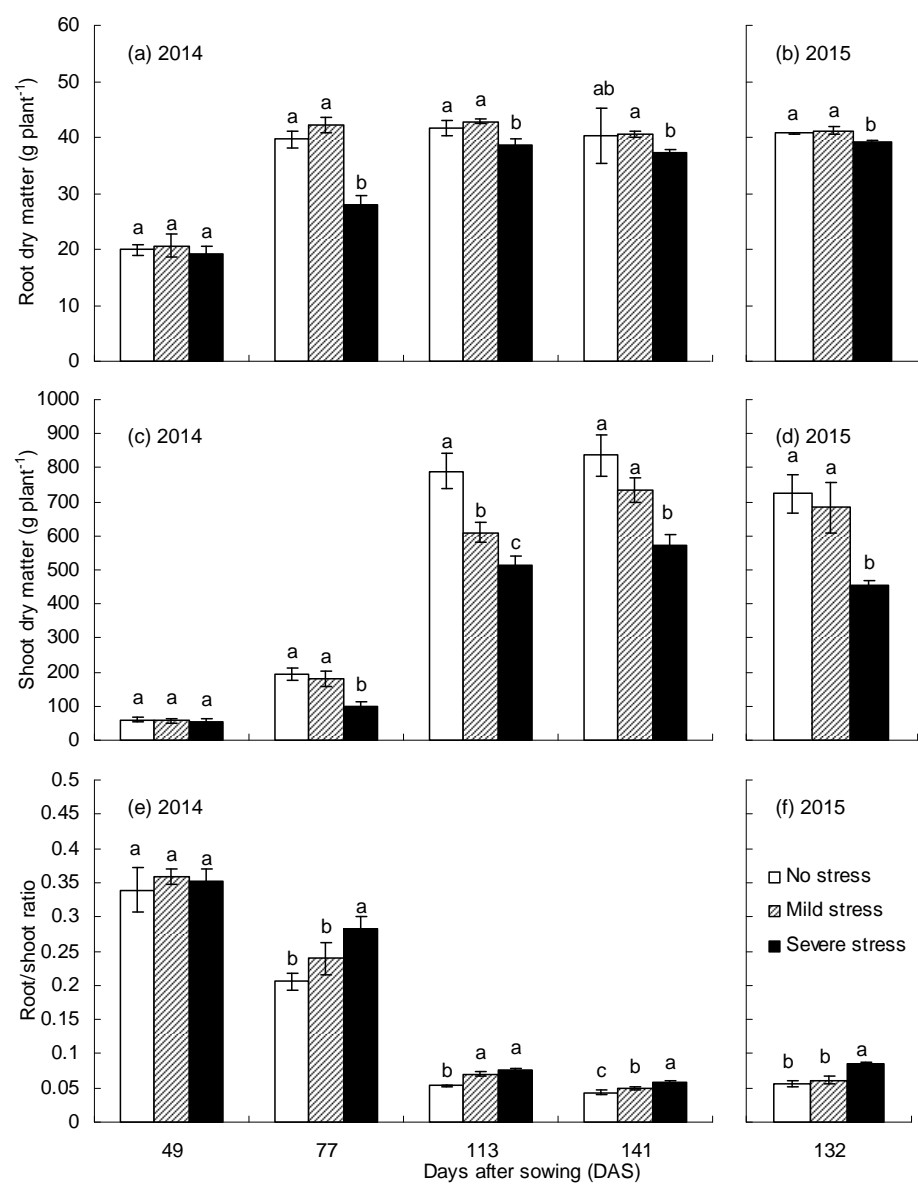


**Fig. 2** Root and shoot dry matters of maize under water stress at different growing

periods in 2014-2015.





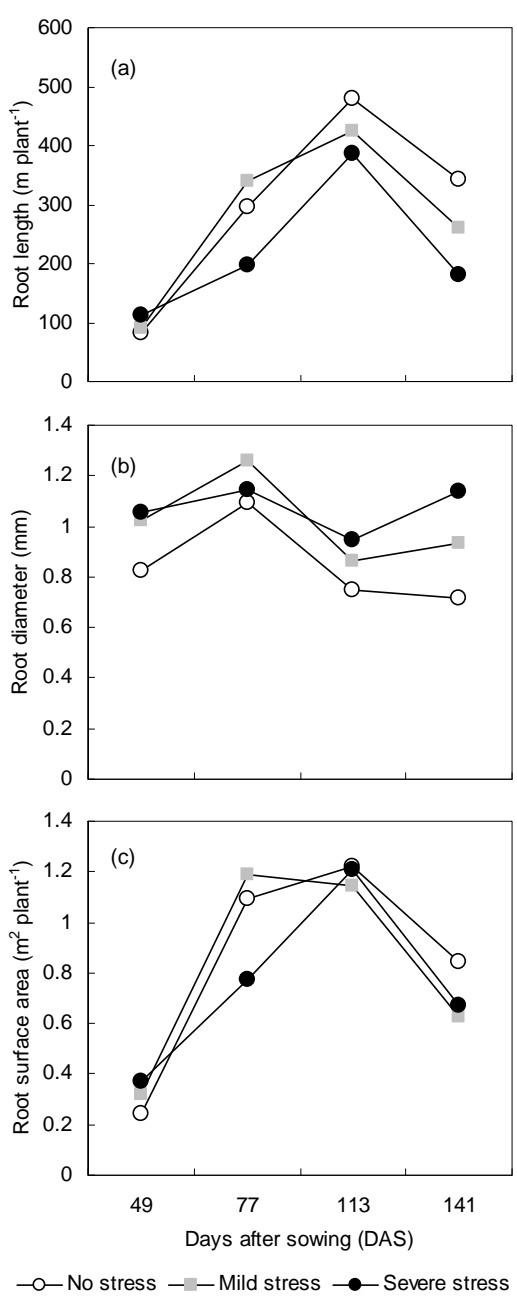


**Fig. 3** Total root length, average diameter and total surface area per plant affected by
water stress in 2014-2015





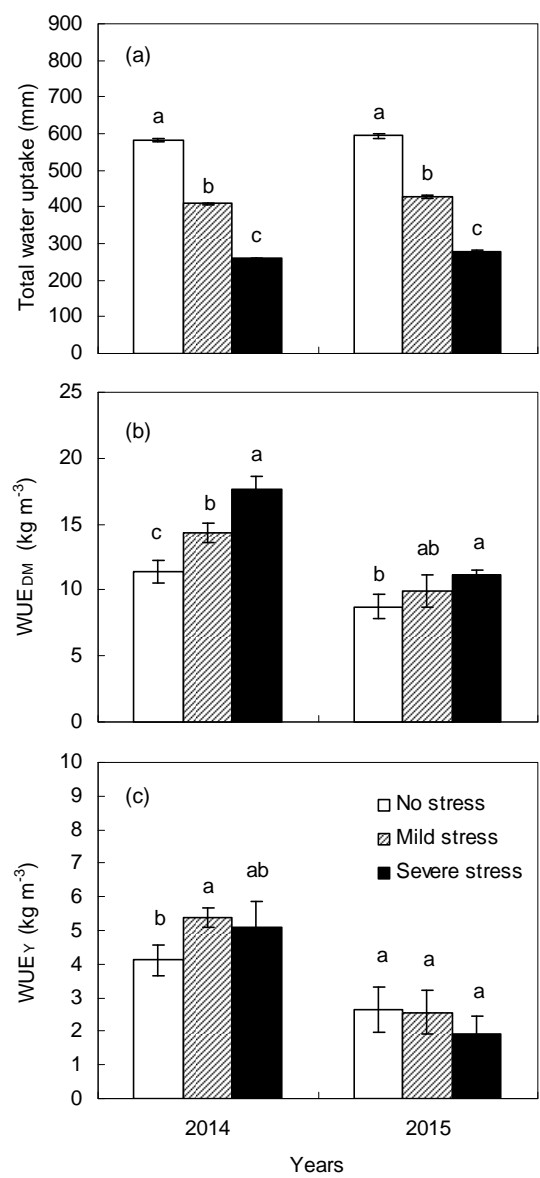


**Fig. 4** Total water uptake (WU) during crop growing season and water use efficiency

for above-ground dry matter (WUE$_{DM}$) and grain yield (WUE$_Y$) under water stress in

2014-2015
