# Peer review of "Morphological plasticity of root growth under mild water stress increases water use efficiency without reducing yield in maize"

_Biogeosciences, 2017_

## Referee Comment (RC1) · Anonymous Referee #1 · 19 Apr 2017

The manuscript focused on yield and water use of rain-fed maize in semi-arid region where the summer drought frequently occurs. For high yielding of maize, the government or farmers are hard to make a decision on whether the irrigation systems is economically necessary, especially under climate change. The paper has a good experimental design by a mobile rain shelter. The results are interesting and valid. Suggestions: (1) provide more informative discussion on the economic analysis of the necessity of irrigation systems. What is current situation, how much area of maize can be irrigated at the time of drought occurrence in rain-fed agriculture, for example at study region, how often was this system used, and how much is the cost. (2) Add the long-term rainfall data and the trend under climate change senarios. This results would

help readers to understand the severity of summer drought.

Detail comments: L78, delete 'Interestingly'. L101-104, give information on the data is averaged from which years. L106-107, since the experiments were conducted under a rain shelter, the detail rainfall in experimental years is not useful, delete. L236, delete 'for'. L295, change 'would increase' to 'increases'. L301, replace 'interesting evidences' by 'more evidences' L442, Table 2, ears plant-1; Kernels ear-1. L445, Figure 1, Capitalize M in legends as 'Maximum air temperature'.

---

## Referee Comment (RC2) · Anonymous Referee #2 · 2 May 2017

General comments: The paper presents an interesting analysis of morphological and physiological response to water stress in a global major crop specie, maize. As water use efficiency (WUE) is a powerful indicator of tradeoff between water transport and dry matter production, understand the variance of WUE with the different degree of water stress is very valuable. The topic is suitable for the journal, and the results provide important instruction for economical irrigation schedule for maize in semi-arid region. However, a minor revision as follows is necessary before the publication: Title & abstract: what is the definition for water stress levels? –no water stress, mild water stress, and severe water stress? Introduction: I think a clear definition of growth stage in a year is necessary ( with a table or figure), which would help us understand the specific means of "V16","R1",etc. Material and method: how does the mobile rain shelter construct and work? What height and how to prevent the possible moisture intrusion from the side? Results: Table 1 provide the actual water supply at three growing periods, what is basis of this quantity? Fig.1 why not consider the solar radiation that may be more related to transpiration and photosynthesis during the growing season? Fig. 2 why selected 49,77,113 and 141 of DAS as time for comparison? Why it is not the same period in 2015? Specific comments: Line 37-38, this sentence as a new point that should be discussed more detailed in Discussion; Line 43, what is critical water-sensitive period? that should be pointed out clearly; Line 78, what is V16,R1? Line 110-111, please consider how to define the water stress, soil water content or actual water supply? Line 115, the word "supplement" is not appropriate for the pot experiments; Line 133-144, what is means of V6,VT,R3, R5? Line 155, what is the number of repetitions for soil samples? Line 212-213 I think this point is not persuasive at this time, it should discuss deeper together with morphological plasticity in Line 259-;

---

## Author Comment (AC1) · 8 Jun 2017

The manuscript focused on yield and water use of rain-fed maize in semi-arid region where the summer drought frequently occurs. For high yielding of maize, the government or farmers are hard to make a decision on whether the irrigation systems is economically necessary, especially under climate change. The paper has a good experimental design by a mobile rain shelter. The results are interesting and valid. Suggestions: (1) provide more informative discussion on the economic analysis of the necessity of irrigation systems. What is current situation, how much area of maize can

be irrigated at the time of drought occurrence in rain-fed agriculture, for example at study region, how often was this system used, and how much is the cost.

Response: In studied region (Liaoning province), almost all maize grow in rain-fed condition (2.4 million ha), covering 73% of total area of grain crops. To reduce the effect of drought stress on maize production, the wells system piping up underground water to irrigate crop is planned recently. The wells need to be 60 to 70 m deep with an average cost of 12,000 Yuan for each. Each well can irrigate 9 to 10 ha of maize. According to our results, only severe drought stress significantly reduced maize yield (up to 50%), which happened less than 5% during 1965 to 2015. Mild drought stress occurs much frequently (28% of years), however, it does not affect maize yield much. Our study suggested that the well system in this rain-fed agriculture might not be economically and ecologically necessary. Other agronomy practices such as intercropping maize with crops requiring less water (e.g. peanut), cultivar selection, adjusting sowing windows (Liu et al., 2013; Lu et al., 2017) and ridge-furrow with plastic film (Dong et al., 2017) are applicable for optimising crop yield and regional sustainability. We added this in the discussion (L312-325).

(2) Add the long-term rainfall data and the trend under climate change senarios. This results would help readers to understand the severity of summer drought.

Response: We added a new figure (Fig. 2) in results for rainfall variations and cumulative frequency from 1965 to 2015 (L199-205).

Detail comments: L78, delete 'Interestingly'.

Response: Done.

L101-104, give information on the data is averaged from which years.

Response: We clarified in the revised text.

L106-107, since the experiments were conducted under a rain shelter, the detail rainfall in experimental years is not useful, delete.

Response: Done.

L236, delete 'for'.

Response: Done.

L295, change 'would increase' to 'increases'.

Response: Done.

L301, replace 'interesting evidences' by 'more evidences'

Response: Done.

L442, Table 2, ears plant-1; Kernels ear-1.

Response: Done.

L445, Figure 1, Capitalize M in legends as 'Maximum air temperature'.

Response: Done.

Please also note the supplement to this comment:
http://www.biogeosciences-discuss.net/bg-2017-103/bg-2017-103-AC1-
supplement.pdf

**Supplement:**

**Morphological plasticity of root growth under mild water stress increases water use efficiency without reducing yield in maize**

Qian Cai[1, 2], Yulong Zhang[1*],Zhanxiang Sun[2*], Jiaming Zheng[2],Wei Bai[2], Yue Zhang[3], Yang Liu[2], Liangshan Feng[2], Chen Feng[2], Zhe Zhang[2], Ning Yang[2], Jochem B. Evers[4], Lizhen Zhang[3]

[1]College of Land and Environment, Shenyang Agricultural University, Shenyang, 110161, Liaoning, China;

[2]Tillage and Cultivation Research Institute, Liaoning Academy of Agricultural Sciences, Shenyang, 110161, Liaoning, China;

[3]College of Resources and Environmental Sciences, China Agricultural University, Beijing, 100193, China.

[4]Wageningen University, Centre for Crop Systems Analysis (CSA), Droevendaalsesteeg 1, 6708 PB Wageningen, The Netherlands

**First author**

Qian Cai, Email: caiqian2005@163.com

**\*Corresponding author**

Yulong Zhang, Email: ylzsau@163.com; Zhanxiang Sun, Email: sunzhanxiang@sohu.com

**Abstract**

Large yield gap exists in rain-fed maize (*Zea mays* L.) production in semi-arid regions, mainly caused by frequent droughts halfway the crop growing period due to uneven distribution of rainfall. It is questionable if irrigation systems are economically required in such a region since total amount of rainfall generally meet the crop requirement. This study aimed to quantitatively determine the effects of water stress during jointing to filling stages on root and shoot growth and the consequences for maize grain yield, above- and below-ground dry matter, water uptake (WU) and water use efficiency (WUE). Pot experiments were conducted in 2014 and 2015 with a mobile rain shelter. The experiments consisted of three treatments: (1) no water stress; (2) mild water stress; and (3) severe water stress. The cumulative frequency for no water stress ( above 500 mm) during maize growing season was 69 % from 1965 to 2015, 28 % for mild water stress (350-450 mm) and 4 % for severe stress (200-300 mm). 
[revised manuscript text omitted]

2014-2015

---

## Author Comment (AC2) · 8 Jun 2017

Anonymous Referee #2 General comments: The paper presents an interesting analysis of morphological and physiological response to water stress in a global major crop specie, maize. As water use efficiency (WUE) is a powerful indicator of tradeoff between water transport and dry matter production, understand the variance of WUE with the different degree of water stress is very valuable. The topic is suitable for the journal, and the results provide important instruction for economical irrigation schedule for maize in semi-arid region. However, a minor revision as follows is necessary before the publication: Title& abstract: what is the definition for water stress levels? –no water stress, mild water stress, and severe water stress?

Response: We took the average rainfall during crop growing season (May to September) as a no water stress control. This amount equals (slight greater than) maize water requirement in this region; mild water stress is 350- 450 mm and severe is 200-300 mm than the average. The possibility of the occurrence of no stress is 69%, mild stress 28% and severe is 4%. We clarified this in the revised text (L117-118).

Introduction: I think a clear definition of growth stage in a year is necessary ( with a table or figure), which would help us understand the specific means of "V16","R1",etc.

Response: We added specific growth stage in the revised text as V16 (with 16 fully expanded leaves) and R1 (silking). In M&M, we cited a reference for international standard of maize growth stages (Adendroth et al., 2011).

Response: Material and method: how does the mobile rain shelter construct and work?

The mobile rain shelter is composed of steel frame and transparent PVC. The mobile rain shelter is built on a mechanical movement track and equipped with a electricity motor to move the shelter by a remote control as shown in Fig. S1. During the time without rain, the shelter is move away from experiments plots and covered only during raining time. We clarified in the revised text (L118-121).

What height and how to prevent the possible moisture intrusion from the side?

Response: The mobile rain shelter is 9 m in width, 30 m in length and 4.5 m in height. The top and both sides of the shelter have PVC transparent board to prevent outside rainfall. There is a water gutter at out side of movement track to drain the rain water. Therefore the rain water intrusion can be avoided. We clarified this in the M&M section (L123-127).

Results: Table 1 provide the actual water supply at three growing periods, what is basis of this quantity?

Response: The actual water supply at three growing periods is based on the maize water requirements and possibility of rainfall distribution. We clarified this in the revised text.

Fig.1 why not consider the solar radiation that may be more related to transpiration and photosynthesis during the growing season?

Response: In Fig. 1, we did not include solar radiation data in two experimental years because the radiation in this region generally is not a limiting factor on maize growth and development. Also we did not use radiation data in the analysis.

Fig. 2 why selected 49,77,113 and 141 of DAS as time for comparison?

Response: Because we started water treatments after maize jointing, and then we measured crops around every 30 days. The measuring times are maize key growth and development stages, representing vegetative period (jointing, R6), tasseling (VT), milk (R3) and dent (R5). We clarified in the revised text (L148-149).

Why it is not the same period in 2015?

Response: We only measured once at final stage in 2015 for biomass of shoot and root.

Specific comments: Line 37-38, this sentence as a new point that should be discussed more detailed in Discussion;

Response: We agreed. We added more discussions.

Line 43, what is critical water sensitive period? that should be pointed out clearly;

Response: We clarified this in the revised text.

Line 78, what is V16,R1?

Response: We clarified this in the revised text.

Line 110-111, please consider how to define the water stress, soil water content or actual water supply?

Response: Good point. We added a new figure with the anomalies of rainfall from 1965 to 2015 and the distribution of its possibility. We clarified this in the revised text.

Line 115, the word "supplement" is not appropriate for the pot experiments;

Response: Revised.

Line 133-144, what is means of V6,VT,R3, R5?

Response: We clarified it.

Line 155, what is the number of repetitions for soil samples?

Response: We clarified in L170 (3 replicates for each treatment).

Line 212-213 I think this point is not persuasive at this time, it should discuss deeper together with morphological plasticity in Line 259.

Response: We agreed. We move this into discussion.

Please also note the supplement to this comment:
http://www.biogeosciences-discuss.net/bg-2017-103/bg-2017-103-AC2-supplement.pdf
* * *
[Figure]

**Fig. 1.** Layout of mobile rain shelter in experimental site

**Supplement:**

**Morphological plasticity of root growth under mild water stress increases water use efficiency without reducing yield in maize**

Qian Cai[1, 2], Yulong Zhang[1*],Zhanxiang Sun[2*], Jiaming Zheng[2],Wei Bai[2], Yue Zhang[3], Yang Liu[2], Liangshan Feng[2], Chen Feng[2], Zhe Zhang[2], Ning Yang[2], Jochem B. Evers[4], Lizhen Zhang[3]

[1]College of Land and Environment, Shenyang Agricultural University, Shenyang, 110161, Liaoning, China;

[2]Tillage and Cultivation Research Institute, Liaoning Academy of Agricultural Sciences, Shenyang, 110161, Liaoning, China;

[3]College of Resources and Environmental Sciences, China Agricultural University, Beijing, 100193, China.

[4]Wageningen University, Centre for Crop Systems Analysis (CSA), Droevendaalsesteeg 1, 6708 PB Wageningen, The Netherlands

**First author**

Qian Cai, Email: caiqian2005@163.com

**\*Corresponding author**

Yulong Zhang, Email: ylzsau@163.com; Zhanxiang Sun, Email: sunzhanxiang@sohu.com

**Abstract**

Large yield gap exists in rain-fed maize (*Zea mays* L.) production in semi-arid regions, mainly caused by frequent droughts halfway the crop growing period due to uneven distribution of rainfall. It is questionable if irrigation systems are economically required in such a region since total amount of rainfall generally meet the crop requirement. This study aimed to quantitatively determine the effects of water stress during jointing to filling stages on root and shoot growth and the consequences for maize grain yield, above- and below-ground dry matter, water uptake (WU) and water use efficiency (WUE). Pot experiments were conducted in 2014 and 2015 with a mobile rain shelter. The experiments consisted of three treatments: (1) no water stress; (2) mild water stress; and (3) severe water stress. The cumulative frequency for no water stress ( above 500 mm) during maize growing season was 69 % from 1965 to 2015, 28 % for mild water stress (350-450 mm) and 4 % for severe stress (200-300 mm). 
[revised manuscript text omitted]

2014-2015

---

## Author Response (AR1)

**Response to editor (bg-2017-103)**

Comments to the Author:
This paper describes an experiment to examine the water stress on corn yield so as to determine the economic feasibility to irrigate the crop with groundwater. It is a concise paper with useful information to the corn producers and researcher as well. However, at its current status, it has not satisfied the level to be accepted to be published in the Journal because of multiple reasons.

(1) There is a great room for improving this paper by intensive English editorial work. There are several cases where the entire section is not well presented in fluent English. Some minor grammatical errors can be found in several locations as well.

**Response:** Thank you for the comments. We thoroughly revised whole text, especially for language and grammar.

(2) Suggest to separate discuss and conclusion so that the value of the paper in science and production can be fully demonstrated. In particular, the scientific value of the paper has been less focused, which should be added.

**Response:** We totally agreed. We separated discuss and conclusions, also revised the text accordingly.

Some specific comments have been attached.
Introduction is generally well presented but need some as commented in the attached file.

**Response:** All comments including in attached PDF file have been revised accordingly.

Material and methods are well described to a proper level.
Result
3.2 Has no information about study period which makes the current data lack value. In addition, the comments from discussion should be carefully addressed.

**Response:** We added the information during study period. The discussion was revised accordingly.

**Non-public comments to the Author:**
This paper describes an experiment to examine the water stress on corn yield so as to determine the economic feasibility to irrigate the crop with groundwater. It is a concise paper with useful information to the corn producers and researcher as well. However, with its current status, it has not satisfied the level to be accepted to be published in the Journal because of multiple reasons.

(1) There is a great room to improve this paper by intensive English editorial work. There are several cases where the entire section is not well presented in fluent English. Some minor grammatical errors can be found in several locations as well.

**Response:** We thoroughly revised whole text according to the comments and especially pay great attention to the English writing.

(2) Suggest to separate discuss and conclusion so that the value of the paper in science and production can be fully demonstrated.

**Response:** We totally agreed. We separated discuss and conclusions, also revised the text accordingly.

Some specific comments have been attached.

**Response:** All comments including in the attached PDF file have been revised accordingly.